# Characterizing the Pathogenicity and Immunogenicity of Simian Retrovirus Subtype 8 (SRV-8) Using SRV-8-Infected Cynomolgus Monkeys

**DOI:** 10.3390/v15071538

**Published:** 2023-07-12

**Authors:** Libing Xu, Yunpeng Yang, Yandong Li, Yong Lu, Changshan Gao, Xinyan Bian, Zongping Liu, Qiang Sun

**Affiliations:** 1Institute of Comparative Medicine, College of Veterinary Medicine, Yangzhou University, Yangzhou 225009, China; 2Jiangsu Co-Innovation Center for Prevention and Control of Important Animal Infectious Diseases and Zoonoses, Yangzhou University, Yangzhou 225009, China; 3Institute of Neuroscience, Center for Excellence in Brain Science and Intelligence Technology, Chinese Academy of Sciences, Shanghai 200031, China

**Keywords:** simian retrovirus, flow cytometry, immune system, RNA-sequencing

## Abstract

Simian retrovirus subtype 8 (SRV-8) infections have been reported in cynomolgus monkeys (*Macaca fascicularis*) in China and America, but its pathogenicity and immunogenicity are rarely reported. In this work, the SRV-8-infected monkeys were identified from the monkeys with anemia, weight loss, and diarrhea. To clarify the impact of SRV-8 infection on cynomolgus monkeys, infected monkeys were divided into five groups according to disease progression. Hematoxylin (HE) staining and viral loads analysis showed that SRV-8 mainly persisted in the intestine and spleen, causing tissue damage. Additionally, the dynamic variations of blood routine indexes, innate and adaptive immunity, and the transcriptomic changes in peripheral blood cells were analyzed during SRV-8 infection. Compared to uninfected animals, red blood cells, hemoglobin, and white blood cells were reduced in SRV-8-infected monkeys. The percentage of immune cell populations was changed after SRV-8 infection. Furthermore, the number of hematopoietic stem cells decreased significantly during the early stages of SRV-8 infection, and returned to normal levels after antibody-mediated viral clearance. Finally, global transcriptomic analysis in PBMCs from SRV-8-infected monkeys revealed distinct gene expression profiles across different disease stages. In summary, SRV-8 infection can cause severe pathogenicity and immune disturbance in cynomolgus monkeys, and it might be responsible for fatal virus-associated immunosuppressive syndrome.

## 1. Introduction

Simian retrovirus (SRV) is a type D retrovirus with a single-stranded, enveloped RNA sequence. The viral reverse transcriptase transcribes RNA to DNA that is then integrated into the host genome. The first SRV, named the Mason Pfizer Monkey Virus (MPMV), was isolated from the mammary carcinoma tissue of rhesus monkeys in 1970 [1]. Since then, various subtypes of SRV (SRV-1 through SRV-8) were isolated from macaques [2].

Like the type D retroviruses, SRV-infected animals develop AIDS-like diseases, e.g., anemia, granulocytopenia, lymphopenia, diarrhea, weight loss, and splenomegaly [3,4,5]. SRV infection disturbs the host’s immune response by suppressing lymphocyte function and decreasing immunoglobulin production [6]. Regarding the pathogenicity of specific SRV subtypes, SRV-1 infection can cause hematological abnormalities, particularly anemia and granulocytopenia, and clinical and lymph node morphological changes [5]. SRV-2 is associated with retroperitoneal fibromatosis and can lead to tumor or other immune-suppressive symptoms [7]. SRV-3, named the Mason Pfizer Monkey Virus (MPMV), shares 83% identity with SRV-1 in the *env* gene and is involved in the induction of a T cell suppressor population [8]. SRV-4 is involved in multifocal lymphoplasmacytic and histiocytic inflammation [9]. SRV-5 can infect a variety of tissues (especially digestive tissues) and impair the function of B cells [10]. Apart from the above-mentioned subtypes, SRV-6 and SRV-7 are rarely mentioned, and SRV-8 was recently identified [11]. Although an SRV-8 epidemic has been reported in cynomolgus monkeys, the pathogenicity and immunogenicity of SRV-8 were not reported.

The advances in RNA sequencing technologies exposed the correlations between retroviruses infection and host antiviral immunity [12,13]. For example, 124 unique ERV loci of human endogenous retroviruses have been identified, showing an IFN-independent signature in human beings [14]. In HIV patients, an activated IFN response pathway can induce the immune response and boost the anti-viral effects during acute infection [15]. SIV infection significantly impacted the immunological profile of non-human primates (NHPs), affecting the expression of genes associated with antivirus response, inflammation, immune activation, and CD4+ T cell death [16]. While these findings expand our understanding of virus–host interactions during retroviral infections, the mechanism of SRV infection in NHPs remains unclear.

Here, we identified SRV-8-infected monkeys exhibiting anemia, weight loss, and persistent unresponsive diarrhea at our facility. These monkeys were categorized into five groups based on the progression of SRV-8 infection. An analysis of HE staining and viral loads revealed that SRV-8 predominantly localized in the intestine and spleen, causing tissue damage. Further investigations demonstrated that SRV-8 infection dynamically affected routine blood indexes as well as innate and adaptive immunity in cynomolgus monkeys. Peripheral blood lymphocytes, mainly B cells and NK cells, are targeted by SRV-8 and exhibit cellular tropism. Additionally, transcriptome sequencing was conducted to examine the distinctive gene expression profile of PBMCs among the different SRV-8-infected monkey groups. Our study characterized the pathogenicity and immunogenicity of SRV-8 in cynomolgus monkeys, suggesting that SRV-8 infection might contribute to the development of a fatal virus-associated immunosuppressive syndrome in NHPs.

## 2. Materials and Methods

### 2.1. Animals

Cynomolgus monkeys were housed at the Non-Human Primate Facility at the Center for Excellence in Brain Science and Intelligent Technology, Chinese Academy of Sciences. All the monkeys were housed in an air-conditioned environment with controlled temperature (22 ± 1 °C), humidity (50% ± 5% RH), 12 h light/12 h dark cycle (lights-on time 07:00 to 19:00), and continuous access to municipal water. All the monkeys were fed periodically with commercial monkey diet (Anmufei, Suzhou, China) twice daily (200 g per monkey at 8:00 am and 15:00 pm), and with fruits and vegetables once daily (100 g per monkey at 10:00 am) to provide essential nutrients and vitamins.

In this work, 120 female cynomolgus monkeys (8–10 years old) with the ill phenotypes of weight loss, anemia, and diarrhea were used. Studying female primates would help us to control the number of SRV-8-infected monkeys, as this virus could be vertically transmitted, and to create a better, specifically pathogen-free (SPF) environment for the monkeys [17]. All these monkeys were housed individually during the study. All experimental procedures were performed according to the ethical guidelines entitled “Non-Human Primate Research Platform Laboratory Animal Assisted Reproduction and Sample Collection Routine Operations” (ION-2019043R02) of the Center for Excellence in Brain Science and Intelligent Technology, Chinese Academy of Sciences.

### 2.2. Verification of SRV Subtype via Polymerase Chain Reaction (PCR) in Euthanized Monkeys

The genomes of 11 tissues (heart, liver, spleen, lung, kidney, uterus, rectum, colon, cecum, sternum, and ilium) that were collected from one euthanized SRV-8-infected monkey (P+V+A+) were extracted using the TIANamp genomic DNA kit (Tiangen, Beijing, China) according to the manufacture’s protocol [18]. The PCR primers (SRV-1/2/4/5/8 -F and -R) were designed to target the conserved envelope (ENV) region of the SRV virus and are listed in Table 1. PCR was carried out with the following cycling conditions: 95 °C for 5 min, followed by 35 cycles at 95 °C for 30 s, 57 °C for 30 s, and 72 °C for 30 s. The resulting PCR products were cloned into the pMD19-T vector (TaKaRa Biomedical Technology, Beijing, China) for sequencing. Sequence similarity was analyzed using the BLAST program (https://blast.ncbi.nlm.nih.gov/Blast.cgi, accessed on 19 October 2019).

### 2.3. Real-Time Quantitative PCR for the Detection of SRV-8 cDNA Copies in Euthanized Monkeys

Total RNA of different tissues and blood samples were extracted using the RNAprep pure Blood Kit (Tiangen, Beijing, China) following the manufacture’s protocol. RNA was converted to cDNA using HiScript II Q RT SuperMix for qPCR (with gDNA wiper) (Vazyme Biotech, Nanjing, China). Real-time PCR was performed using AceQ qPCR SYBR^®^ Green Master Mix (Vazyme, Q111-02/03) [19,20]. Quantitative PCR was carried out with a LightCycler^®^ 480 II PCR system (Roche, Basel, Switzerland) with the following cycling conditions: 95 °C for 5 min, 45 cycles at 95 °C for 10 s, 60 °C for 10 s, and 72 °C for 10 s. The relative expression levels of these selected genes were calculated using normalization to *GAPDH* [21].

### 2.4. Detection of SRV-8 Antibodies in Euthanized Monkeys

The blood samples of SRV-8-infected monkeys were collected in tubes without anticoagulants. The plasma was collected after centrifugation at 1000× *g* for 20 min at 4 °C and used for ELISA experiments. In total, 50 μL of the prepared plasma sample was used to detect the SRV-8 antibody according to the standard method of the antibody detection kit (VRL Laboratories, Suzhou, China).

### 2.5. Haematoxylin and Eosin (HE) Staining

The spleen, caecum, colon, and rectum of wild-type and SRV-8-infected monkeys were collected and fixed in 10% formalin and embedded in paraffin. Then, 5 μm paraffin-embedded sections were used for HE staining according to the standard protocol [22].

### 2.6. Collection of PBMCs for CPE Formation Analysis and Immunofluorescent Assay (IFA)

Peripheral blood mononuclear cells (PBMCs) were isolated from SRV-8-infected monkeys containing SRV-8 provirus using ficoll gradient centrifugation. The collected PBMCs were washed in phosphate-buffered saline (PBS), and aliquoted to 10^6^ cells per/mL [23]. Then, 5 × 10^5^ PBMCs were mixed with 2 × 10^5^ Raji cells (a Burkitt’s lymphoma B-cell line) and co-cultured in RPMI-1640 medium supplemented with 10% fetal bovine serum, 2-mercapto-ethanol (50 μM), sodium pyruvate (1 mM), penicillin (100 IU/mL), streptomycin (100 μg/mL), concanavalin A (10 μg/mL), and recombinant interleukin-2 (200 U/mL) (Shionogi and Co., Ltd., Osaka, Japan). The viral cytopathic effect (CPE) was monitored as previously reported [24]. When CPE was detected obviously in Raji cells, the cells were fixed in ice-cold 2.5% glutaraldehyde in 0.1 M sodium cacodylate buffer (pH 7.5) and washed with 0.1 M cacodylate buffer containing 0.1 M sucrose. The fixed cells were embedded in Epon 812 resin. The ultra-thin sections were stained with lead citrate/uranium acetate and analyzed using an electron microscope (Hitachi, Tokyo, Japan).

For immunofluorescent assay (IFA) analysis, SRV-infected Raji cells were collected at 450× *g* for 5 min, and washed twice with cold PBS. Optimal cell numbers were placed in each well of IFA slides. Slides were dried and fixed using ice-cold acetone (Fisher Scientific, Pittsburgh, PA, USA). Cells were stained with the primary antibody (SRV-8-positive animal sera pool collected from VRL Laboratories, Suzhou, China) at 37 °C for 30 min, followed by incubation with a FITC labeled goat anti-human IgG conjugate (0.1% Evans Blue, Bion, Des Plaines, IL, USA). Slides were mounted using mounting medium (Bion, Des Plaines, IL, USA) and observed under a Nikon ECLIPSE 55i microscope [25].

### 2.7. Detection of Viremia, Proviral DNA, and Antibodies of SRV-8 in 28 Live Monkeys

The blood samples collected from wild-type and SRV-8-infected monkeys were used to detect the viremia (RNA), proviral DNA, and/or the antibodies of SRV-8 by using qRT-PCR, RT-PCR, and ELISA analysis according to the standard protocol of VRL Laboratories (VRL-China). The primers are listed in Table 1, and the values of sensitivity and specificity in the Elisa kit were 99.1% and 99.6%, respectively.

As for the ELISA analysis, 100 μL of monkey sera, diluted 1:50 in sample diluent, was added to each well and the plate was incubated for 0.5 h at 37 °C in a moist chamber. After three washes with 350 μL wash buffer for 3 min, 100 μL per well of HRP conjugate was added and incubated for 0.5 h at 37 °C in a moist chamber. After three washes with 350 μL wash buffer, 100 μL of HRP substrate was added to each well and the plate was incubated for 20  ±  5 min at room temperature in the dark. The reaction was stopped with 100 μL stop solution. The optical density (OD) was measured using an automatic micro-ELISA (Anthos 2020, Cambridge, UK) at a wavelength of 405 nm.

### 2.8. Routine Blood Examination

The blood samples of wild-type and SRV-8-infected cynomolgus monkeys were collected and analyzed with a chemistry analyzer (Catalyst One, Westbrook, ME, USA).

### 2.9. Flow Cytometry Analysis

To study the effect of SRV-8 infection on the variation of immune cells in innate and adaptive immunity, single-cell suspensions of PBMCs were prepared according to the manufacturer’s protocol [23], and then subjected to flow cytometry staining analysis by using a NovoCyte Advanteon (ACEA Biosciences, San Diego, CA, USA). The results were analyzed with NovoExpress software (ACEA Biosciences, San Diego, CA, USA). In detail, the immune cells were resuspended in PBS buffer containing 2% FBS and incubated with the cell-specific monoclonal antibodies (mAb) for 30 min at 4 °C. The cells were washed twice with PBS buffer before FACS analysis. Different cell subsets were stained using human-specific antibodies that cross-reacted with monkey antigens. The following combinations of PE-Cy7/APC/FITC/PE/APC-Cy7-labeled monoclonal antibodies (mAb) were used for multi-color FACS analysis: APC-Cy7/FITC (anti-CD3^+^/4^+^; reactive with helper T cells); APC-Cy7/APC (anti- CD3^+^/8^+^; reactive with cytotoxic T cells); PE-Cy7 (anti-CD20^+^; reactive with B cells); PE/APC (anti-CD56^+^/16^+^; reactive with NK cells); and APC/FITC (anti-CD45^+^/34^+^; reactive with hematopoietic stem cells). The panel and gating strategy made clear separation of different cell subsets. All monoclonal antibodies were purchased from BioLegend (BioLegend, San Diego, CA, USA). Cells incubated with FITC/PE/APC/PE-Cy7/APC-Cy7-anti-mouse Ig (Becton Dickinson, Sunnyvale, CA, USA) were used as controls for each analysis.

For cellular immune response, an in vitro phagocytosis assay was performed as previously described [26]. Briefly, PBMCs collected from the wild-type and five SRV-8-infected monkey groups were grown in 96-well plates (1 × 10^5^ cells/well) and then incubated with 2 µm diameter carboxylate-modified polystyrene fluorescent beads (L4530, St. Louis, MO, USA) in 5% CO_2_ incubator at 37 °C for 1 h. After incubation, the cells were washed with ice-cold fluorescence-activated cell sorting (FACS) buffer (phosphate-buffered saline (PBS), Invitrogen, 14190-169). The suspended cells were analyzed by using FITC channel in a NovoCyte Advanteon (ACEA Biosciences, San Diego, CA, USA). The result was analyzed using NovoExpress software (ACEA Biosciences, San Diego, CA, USA).

The effect of SRV-8 infection on the respiratory burst activity of neutrophils was analyzed as previously reported [27]. In brief, phorbol-12-myristate-13-acetate (PMA) was used to stimulate neutrophils to produce reactive oxides, which could induce the oxidation of dihydrorhodamine to rhodamine 123 and then emit fluorescence. According to the protocol of neutrophils respiratory burst quantitative assay kit (Absin Bioscience, Shanghai, China), 50 μL of blood samples were mixed with 50 μL PMA and incubated at 37 °C for 15 min. Subsequently, 20 μL of dihydrorhodamine was added to the mixture and incubated at 37 °C for 5 min in the dark. Then, 1 mL of diluted hemolysin was added to the mixture at room temperature and hemolyzed for 15 min. The mixture was washed twice with PBS buffer and centrifuged at 1500 rpm for 5 min. Finally, the collected cells were resuspended in 0.5 mL PBS buffer for flow cytometry analysis.

### 2.10. Transcriptome Sequencing Analysis

Total RNA of PBMCs was isolated using TRIzol reagent (TaKaRa Biomedical Technology, Beijing, China), according to the manufacturer’s instructions. Genomic DNA was removed using DNase (TRANSGEN Biotech, Beijing, China). The purity of RNA was assessed by using NanoDrop 2000 (NanoDrop Technologies, Pittsfield, MA, USA). The integrity of RNA samples was detected with agarose gel electrophoresis. Then, the RNA samples were used for the library construction as previously reported [28]. Each library was loaded into one lane of the Illumina NovaSeq for 2 × 150 bps pair-end (PE) sequencing.

### 2.11. Identification of the Differentially Expressed Genes (DEGs)

The raw reads were cleaned by removing low-quality reads and adaptor sequences. The clean reads were mapped to the Macaca fascicularis genome using HISAT2 software [29]. Mapped clean reads were assembled into transcripts with StringTie (Version 1.3.3b) [30]. Gene expression was expressed as transcripts per million reads (TPM) and was calculated using RSEM (Version 1.3.1) [31] with default settings. Differential expression analysis was performed using DESeq2 (Version 1.24.0) [32]. Genes with log2 fold changes (FC) ≥ 4 and *p* value < 0.05 were defined as DEGs. The filtered genes were used for further analysis.

### 2.12. DEG-Based KEGG Enrichment Analysis

DEG-based KEGG enrichment analyses were conducted to determine the KEGG pathways enriched in the DEGs. KEGG enrichment analysis was carried out with R script based on the KEGG database (http://www.genome.jp/kegg/, accessed on 5 January 2023, Version 2017.08) with Fisher’s exact test [33]. The *p* values produced from the KEGG enrichment analysis were corrected to control the false discovery rate (FDR) using the Benjamini and Hochberg (BH) method [34]. For DEG-based KEGG enrichment analysis, the pathways with FDR < 0.05 were considered to be significantly enriched pathways (SEPs).

### 2.13. Statistical Analysis

All statistical analyses were calculated using Graphpad software 8.0.2 (https://www.graphpad.com/scientific-software/prism/, accessed on 23 September 2022). Statistical analysis data were presented as mean ± SEM. The statistical significance of different parameters between the two monkey groups was analyzed using student *t*-test (unpaired, two-tailed).

## 3. Results

### 3.1. Identification and Stratification of SRV-8-Infected Cynomolgus Monkeys

Between October 2019 and September 2020, we identified 120 female cynomolgus monkeys in our facility who exhibited at least one of the following disease features: weight loss, anemia, and diarrhea (Appendix A). Given the similarity of these symptoms to those observed in simian retrovirus (SRV)-infected monkeys [17], we hypothesized that these monkeys might be infected with SRV. To test this hypothesis, we performed polymerase chain reaction (PCR) using specific primer pairs targeting the 198 bp envelope (*ENV*) region of different SRV subtypes (Figure 1A). As shown in Figure 1B, the proviral genome of SRV-8 was detected in 8 monkeys (range of P^+^V^−^A^−^, P^+^V^+^A^−^, P^+^V^+^A^+^, and P^+^V^−^A^+^), while other SRV subtypes (SRV-1/2/4/5) were not detected in the colony. Subsequently, the PCR products were cloned into the pMD19-T vector for sequencing, and all the sequences showed a high similarity to the *ENV* sequence of SRV-8 (Figure 1C). Furthermore, the prevalence of SRV-8 infection was assessed by performing PCR on all 120 ill monkeys. As shown in Appendix A, 69.2% (83/120) of the ill monkeys tested positive for SRV-8 proviral DNA. Thus, SRV-8 was the predominant strain circulating in our facility, and it likely contributed to the observed disease phenotypes in the affected monkeys.

SRV infection could be classified in three stages: the formation of SRV provirus, the release of viral particles, and the production of SRV-specific antibodies. In this study, SRV-8 provirus, viral particles, and antibody responses were measured in all ill monkeys (Appendix A). Based on these results, the SRV-8-infected monkeys were stratified into five groups (Figure 1D). The number distribution of the total 120 monkeys is shown in Figure 1E. Taking the changeable status of virus development progress during 2 months into consideration, we finally chose the stable 28 monkeys to complete further research. The percentage of the 28 monkeys is shown in Figure 1F: (1) positive for provirus (P) formation but negative for viral particles (V) release and specific antibody (A) responses (P^+^V^−^A^−^, *n* = 3); (2) positive for P and V, but negative for A (P^+^V^+^A^−^, *n* = 6); (3) positive P, V, and A (P^+^V^+^A^+^, *n* = 2); (4) positive for P and A, but negative for V (P^+^V^−^A^+^, *n* = 6); and (5) positive for A but negative for P and V (P^−^V^−^A^+^, *n* = 6). Additionally, uninfected age-matched healthy monkey (P^−^V^−^A^−^, *n* = 5) were included as negative controls.

### 3.2. Tissue Specificity and Pathology of SRV-8 Infection

To understand SRV-8 tissue tropism in cynomolgus monkeys, we analyzed the relative copy number of proviral DNA and cDNA in different tissues collected from one euthanized SRV-8-infected monkey (P^+^V^+^A^+^). SRV-8 was specifically enriched in the caecum, colon, rectum, and spleen (Figure 2A). HE staining showed significant histological lesions on the spleen, caecum, colon, and rectum of SRV-8-infected monkeys, when compared with healthy controls (Figure 2B). Severe leukopenia and significant macrophage infiltration were observed in the spleen of SRV-8-infected monkeys, leading to the disruption of red and white pulp regions. In the large intestine (caecum, colon, and rectum), exfoliated cells or gland atrophy accompanied by reactive hyperplasia of lymphoid follicles were observed in SRV-8-infected monkeys (Figure 2B). To evaluate the cytopathic effect (CPE) of SRV-8, peripheral blood mononuclear cells (PBMCs) from SRV-8-infected monkeys (P^+^V^+^A^+^) were co-cultured with human B Raji cells. As depicted in Appendix A, the characteristic CPE associated with simian retroviruses was observed at days 3, 6, 8, and 13 post-infection, indicating that SRV-8 could infect and replicate in B Raji cells. An immunofluorescent assay using SRV-8 positive animal sera staining confirmed the presence of SRV-8 (Figure 2C). Three virus strains were characterized. Additionally, 198 bp ENV was identified from PCR products by collecting human Raji B cells after being co-cultured with SRV-8 positive animal sera as templates (Figure 2D).

### 3.3. The Health Status and Cellular Immune Responses in SRV-8-Infected Monkeys

To assess the health status of SRV-8-infected monkeys, we analyzed routine blood indexes in six different monkey groups (Figure 1D). Compared to the healthy controls (P^−^V^−^A^−^), the SRV-8-infected monkey groups P^+^V^−^A^−^, P^+^V^+^A^−^, and P^+^V^−^A^+^ exhibited decreased red blood cell (RBC) counts and hemoglobin (HGB) content (Figure 3A). Additionally, the monkey groups P^+^V^+^A^−^, P^+^V^+^A^+^, P^+^V^−^A^+^, and P^−^V^−^A^+^ displayed decreased mean corpuscular volume (MCV) and red blood cell distribution width standard deviation (RDW-SD), while the P^+^V^−^A^−^ group showed increased values for these parameters (Figure 3A). Furthermore, the monkey groups P^+^V^+^A^−^, P^+^V^+^A^+^, and P^−^V^−^A^+^ showed decreased white blood cell (WBC) and lymphocyte (LYMPH) counts (Figure 3B). Due to significant changes in immune cell populations among SRV-8-infected monkeys, we next conducted a flow cytometry analysis to assess the variations in specific immune cell subsets, e.g., helper T cells, cytotoxic T cells, B cells, and natural killer cells (Appendix A). The percentage of helper T cells increased in the monkey group P^+^V^+^A^+^, whereas the percentage of cytotoxic T cells exhibited the opposite trend (Figure 3C). In contrast, the percentage of B cells and natural killer cells decreased in the monkey groups P^+^V^−^A^−^ and P^+^V^+^A^−^, but returned to normal levels in seropositive animals (P^+^V^+^A^+^, P^+^V^−^A^+^, and P^−^V^−^A^+^) (Figure 3C). As previously reported, SRV infection has been associated with direct or indirect damage to CD34^+^ progenitor cells in cynomolgus macaques [18]. Consistent with these findings, we showed a significant decrease in the percentage of hematopoietic stem cells (HSCs) in seronegative infected monkeys (P^+^V^+^A^−^), which was restored in the seropositive groups (P^+^V^+^A^+^, P^+^V^−^A^+^, and P^−^V^−^A^+^, Figure 3D). These results suggest that the released SRV-8 virus may infect HSCs in cynomolgus monkeys, leading to cell death and reduced proliferation. However, this seems to be reversed through antibody-mediated viral clearance. Respiratory burst plays a crucial role in innate immunity by facilitating microbe clearance via phagocytic cells [19]. As depicted in Figure 3E, among the six monkey groups, the respiratory burst activity was highest in the P^+^V^−^A^−^ group. It shows that a respiratory burst of neutrophils is triggered at early infection stages. However, neutrophil phagocytic capability was diminished through antibody-mediated viral clearance (Figure 3E).

### 3.4. Gene Expression Profile of PBMCs in SRV-8-Infected Monkeys

To investigate transcriptomic changes in PBMCs induced by SRV-8 infection, we next conducted transcriptome sequence analysis and compared the gene expression profiles across the six study groups (Figure 4A). Compared to the healthy monkey group (P^−^V^−^A^−^), SRV-8-infected monkey groups exhibited unique gene expression profiles (Figure 4A). Cluster analysis revealed that seropositive animals (P^+^V^+^A^+^, P^+^V^−^A^+^, and P^−^V^−^A^+^) displayed close phylogenetic relationships with the healthy controls, particularly the convalescent group P^−^V^−^A^+^ (Figure 4A). The number of differentially expressed genes (DEGs) (fold change ≥ 4, *p* < 0.05) in the monkey groups P^+^V^−^A^−^, P^+^V^+^A^−^, P^+^V^+^A^+^, P^+^V^−^A^+^, and P^−^V^−^A^+^ were 141, 168, 99, 88, and 59, respectively (Figure 4B, Appendix A). Among these groups, the number of DEGs significantly decreased in the P^−^V^−^A^+^ group, suggesting a return to homeostasis after the antibody-mediated viral clearance of the SRV-8 provirus and virus.

### 3.5. KEGG Enrichment Analyses of DEGs in SRV-8-Infected Monkeys

To explore the function of the DEGs SRV-8-infected monkeys across different disease stages, we next performed a KEGG enrichment analyses (Figure 5). During the early disease stages (P^+^V^−^A^−^), SRV-8 infection induced the expression of genes related to pathogen-infection-related diseases (pertussis, malaria, chagas disease, African trypanosomiasis, tuberculosis, leishmaniasis, coronavirus disease COVID-19, and toxoplasmosis). Meanwhile, genes related to cancer (basal cell carcinoma, pathways in cancer, thyroid cancer, breast cancer, and gastric cancer), carbohydrate metabolism (pentose phosphate pathway, fructose and mannose metabolism, and glycolysis/gluconeogenesis), and cell proliferation and division (Wnt and Hippo signaling pathway) were downregulated (Figure 5A). The expression of genes related to immunity (complement and coagulation cascades and the B-cell-receptor signaling pathway) were also downregulated at this early infection stage (P^+^V^−^A^−^) (Figure 5A), suggesting that SRV-8 infection caused the damage of antimicrobial substances in body fluids before the formation of specific immune responses.

During the peak of viremia (P^+^V^+^A^−^ group), we observed the upregulation of genes related to pathogen-infection-related diseases (influenza A, malaria, coronavirus disease COVID-19, measles, and African trypanosomiasis); genes related to immune modulation (viral protein interaction with cytokine and cytokine receptors, cytokine–cytokine receptor interaction, the chemokine signaling pathway, and the Toll-like receptor signaling pathway); inflammatory responses (the IL-17 signaling pathway); and phagocytosis (Fc gamma R-mediated phagocytosis) (Figure 5B). Interestingly, genes related to neuron activity and communications (dopaminergic synapse, retrograde endocannabinoid signaling, the synaptic vesicle cycle, neuroactive ligand–receptor interaction, and the calcium signaling pathway) were downregulated (Figure 5B), indicating that the released SRV-8 virus might cross the blood–brain barrier and affect the nervous system [35]. This assumption was supported by the downregulation of genes related to nervous diseases, i.e., nicotine addiction, amphetamine addiction, and morphine addiction (Figure 5B).

Seropositive animals (P^+^V^+^A^+^) upregulated the expression of genes related to neurodegenerative diseases (Parkinson’s disease, amyotrophic lateral sclerosis, Alzheimer’s disease, and pathways of neurodegeneration multiple diseases), while the expression of genes related to amino acid metabolism was downregulated (Figure 5C). Moreover, the expression of genes related to immunity (complement and coagulation cascades and the B cell receptor signaling pathway) were still downregulated in group P^+^V^+^A^+^ (Figure 5C). Thus, although the SRV-8-specific antibodies were secreted in this monkey group, the function of the immune system was still inhibited at this stage.

During disease resolution (P^+^V^−^A^+^), the expression of genes related to immunity (the prolactin signaling pathway and intestinal immune network for IgA production) were upregulated, which is consistent with viral clearance (Figure 5D). The expression of genes related to pathogen-infection-related diseases (chagas disease, legionellosis, amoebiasis, toxoplasmosis, African trypanosomiasis, malaria, measles, and influenza A) were downregulated. Moreover, the expression of genes related to neuron communications (cholinergic synapse, synaptic vesicle cycle, and neuroactive ligand–receptor interaction) was also upregulated, suggesting a recovery of the nervous system function (Figure 5D).

With the clearance of provirus integration and viremia transmission (P^−^V^−^A^+^), the genes related to carbohydrate metabolism (thiamine metabolism, glycolysis/gluconeogenesis, and galactose metabolism), amino acid metabolism, nitrogen metabolism, and neuron communications (neuroactive ligand–receptor interaction) were upregulated, while the genes related to the biosynthesis and secretion of steroid hormones (steroid hormone biosynthesis, ovarian steroidogenesis, cortisol synthesis and secretion, and Aldosterone synthesis and secretion) and nervous synapse (GABAergic synapse, cholinergic synapse, serotonergic synapse, and dopaminergic synapse) were downregulated (Figure 5E).

## 4. Discussion

Simian retrovirus type D is enzootic in cynomolgus monkeys (*Macaca fascicularis*) and rhesus macaques (*Macaca mulatta*), and SRV-1–5 have been reported among the reared monkeys while SRV-6 and SRV-7 are rarely documented as it is commonly detected in wild monkeys [11,25]. Until now, the pathogenicity and immunogenicity of SRV-8 in NHPs was rarely reported. Here, we identified the SRV-8-infected cynomolgus monkeys and explored the influence of SRV-8 infection on the tissue-specific pathology and modulation of the immune system. Stratifying SRV-8-infected animals across five groups based on symptoms and serological responses, we explored the transcriptional changes in the peripheral blood across the different disease stages of SRV-8-infection.

SRV-8 was firstly identified in the cynomolgus monkeys in China and the USA, and all infected monkeys appeared healthy and asymptomatic [11]. Contrastingly, here, the majority of infected animals (69.2%) were symptomatic, presenting at least one of the following symptoms: weight loss, anemia, and diarrhea (Appendix A). Assisted with the detection of SRV-8 provirus and virus particles, it was the causative agent for the above-mentioned diseases.

Previous reports showed that SRV infection can bring significant histologic tissue lesions to different organs [36]. However, no relevant studies were reported on SRV-8-infected animals. SRV-8 was more phylogenetically and antigenically related to SRV-4 [11]; here, we compared the histologic lesions of SRV-8-infected monkeys with that of the SRV-4-infected macaques [37]. Just like SRV-4, SRV-8 showed broad tissue tropism in cynomolgus monkeys, targeting digestive organs (caecum, colon, and rectum) with significant histologic lesions (Figure 2A,B).

SRV-4 infection in both the Japanese macaques and a humanized mouse model can lead to hemorrhagic anemia, featuring a reduction in red blood cell numbers, hematocrit, and hemoglobin [37,38]. Similarly, we observed a reduction of red blood cell (RBC) number and hemoglobin content (HGB) in the SRV-8-infected monkeys (groups P^+^V^−^A^−^, P^+^V^+^A^−^, and P^+^V^−^A^+^, Figure 3A). As to the number of white blood cells (WBC), a significant reduction was observed in both the SRV-4- and SRV-8-infected monkeys (Figure 3B) [37], suggestive of immunodeficiency. Additionally, our results showed the percentage of B and NK cells were decreased in seronegative animals (groups P^+^V^−^A^−^ and P^+^V^+^A^−^, Figure 3C). Numerous retroviruses, including AEV, FLV, M-MuLV, and HTLV-1, can infect HSCs, deregulating normal hematopoiesis and sometimes developing into leukemia/lymphoma [39]. Consistent with these results, SRV-8-infected HSCs of cynomolgus monkeys (Figure 3D). Importantly, the transcriptomic changes in HSCs was quite similar to that of the B and NK cells of SRV-8-infected monkeys, suggesting that the decreased B cells and NK cells were derived from the reduction of HSCs. We also observed increased respiratory burst activity during early infection stages (P^+^V^−^A^−^ group) which implied a strong neutrophil antimicrobial response (Figure 3E).

SRV-8 infection elicited dynamic changes in various host genes [40]. The pathobiological mechanism of the diseases induced by SRV-8 infection is still unknown. To explore the pathological mechanism of SRV-8 infection, we compared the transcriptional changes of PBMCs across different disease stages. Consistent with the dynamics of acute infection and resolution, the number of DEGs increased at the early stage (monkey groups P^+^V^−^A^−^ and P^+^V^+^A^−^) and then decreased with the production of SRV-8-specific antibodies and viral clearance (Figure 4B). Immunodeficiency occurred frequently in SRV-infected animals [5,36]. Similarly, in line with a decrease in B and NK cells during early disease stages (Figure 3C), the expression of genes related to immunity were also decreased in susceptible animals (Figure 5A,B). Disease progression was marked by the increased expression of genes related to immunity, and most infected monkeys were resistant to infection, developing immunity and clearing the viral pathogen (Figure 5D). The metabolism of carbohydrates and amino acids is important for the growth, proliferation, and effector functions of immune cells [41,42]. Consistently, the expression of genes related to carbohydrates and amino acid metabolism decreased at the early stages of SRV-8 infection (monkey groups P^+^V^−^A^−^ and P^+^V^+^A^+^) (Figure 5A,C), but were restored after viral clearance (Figure 5E). Compared to the P^−^V^−^A^−^ group, genes related to host homeostasis (*ALPL*, *GRM2*, *ALAS2*, and *ABCB1*) and immune cell functions (*IL12A, LIFR, CXCL11,* and *MSR*) were upregulated in the P^−^V^−^A^+^ group (Appendix A).

## 5. Conclusions

In this study, SRV-8 was identified as the only serotype of simian retroviruses in cynomolgus monkeys, and the pathogenicity and tissue distribution of SRV-8 was investigated. By stratifying SRV-8-infected monkeys by disease stage, we explored the dynamic changes in blood routine indexes, innate and adaptive immunity, and the transcriptome of PBMCs in SRV-8-infected cynomolgus monkeys. Altogether, this study contributes to a more comprehensive understanding of SRV-8 pathogenesis in NHPs. Notwithstanding its impact, this standing presents some limitations; (1) the study only included female cynomolgus monkeys, limiting its extrapolation to males; (2) despite exploring the expression profile of peripheral PBMCs in SRV-8-infected monkeys, the underlying pathobiological mechanisms of SRV-8 infection remain unclear and require further investigation.

## Figures and Tables

**Figure 1 viruses-15-01538-f001:**
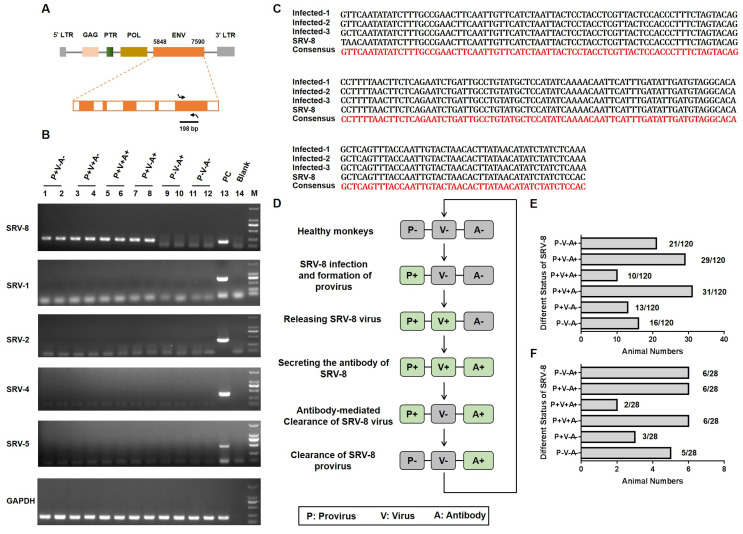
Identification and classification of SRV-8-infected cynomolgus monkeys. (**A**) Schematic diagram illustrating the amplification of 198 bp sequence in the envelope region of the SRV virus. Arrows indicate the direction of primer design. (**B**) PCR-based identification of SRV-1/2/4/5/8 proviral genomes in 12 monkeys. GAPDH was used as the reference gene. (**C**) Sequence alignment of three PCR-amplified sequences with the corresponding sequence of SRV-8. (**D**) Schematic diagram illustrating the different SRV-8-infected monkey groups and the progression of SVR-8 infection in cynomolgus monkeys. (**E**) The number distribution of total 120 SRV-8-related animals in six groups. In total, 16 monkeys were detected as the healthy group, with P^−^V^−^A^−^ phenotype; 104 monkeys were tested as SRV-8-infected animals. (**F**) The number distribution of 28 SRV-8-related animals for further researches. Five animals were detected as being in the healthy group, with P^−^V^−^A^−^ phenotype. A total of 23 animals were detected as being infected with SRV-8, including 6 P^−^V^−^A^+^ animals, 6 P^+^V^−^A^+^ animals, 2 P^+^V^+^A^+^ animals, 6 P^+^V^+^A^−^ animals, and 3 P^+^V^−^A^−^ animals.

**Figure 2 viruses-15-01538-f002:**
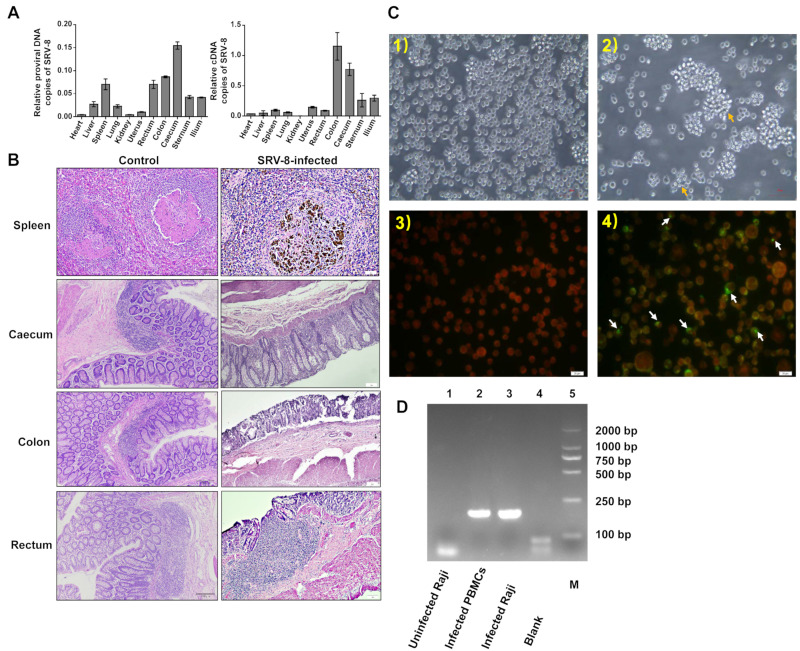
Tissue specificity and pathology of SRV-8 infection. (**A**) Viral loads analysis of SRV-8 in different tissues of SRV-8-infected monkeys. (**B**) Histopathogical changes of the spleen, caecum, colon, and rectum in SRV-8-infected monkeys. (**C**) Human B Raji cells were co-cultured with cynomolgus monkey sera in IFA. White arrows point out the SRV-8-infected Human Raji B cells. (**1**) Uninfected Raji cells. (**2**) Infected Raji cells showing syncytia formation (arrows). (**3**) Uninfected Raji cells stained with sera from SRV-8-positive animals and FITC-conjugated anti-human IgG showing red color under fluorescent microscopy. (**4**) Infected cells were observed as yellow-green in color with a red background for negative cells using immunofluorescent staining. (**D**) PCR identification of 198 bp ENV in uninfected Raji cells, infected donor PBMCs, and infected Raji cells during IFA tests. M stands for Marker, 2000 bp.

**Figure 3 viruses-15-01538-f003:**
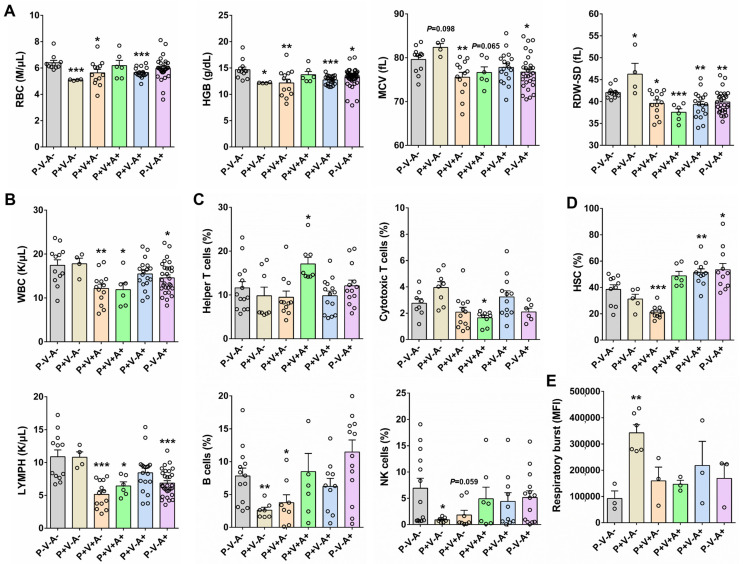
Analysis of routine blood indexes and immune cell subsets in SRV-8-infected monkeys. (**A**) The comparison of RBC, HGB, MCV, and RDW-SD in six monkey groups. (**B**) The comparison of WBC and LYMPH in six monkey groups. (**C**) Flow cytometry analysis of immune cell subsets in six monkey groups. (**D**) The comparison of HSC in six monkey groups. (**E**) The comparison of respiratory burst activity in six monkey groups. **p* < 0.05, ***p* < 0.01, ****p* < 0.001. Each circle represents a sample in this research.

**Figure 4 viruses-15-01538-f004:**
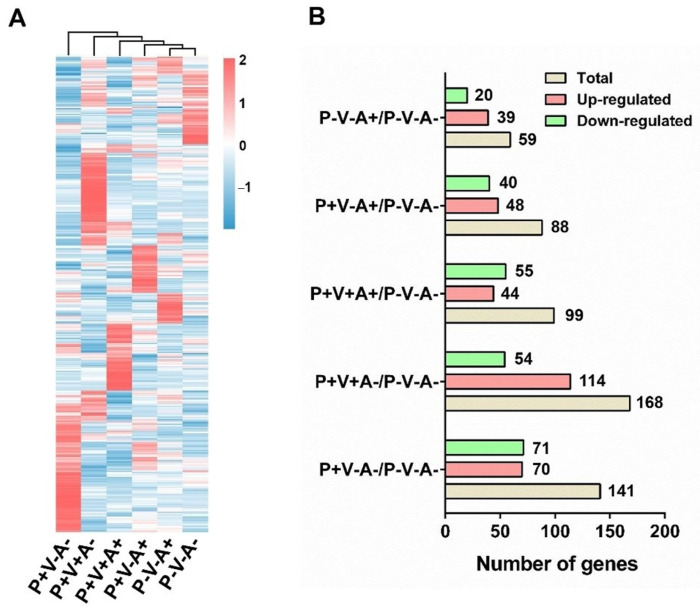
Analysis of changes in PBMC transcriptome in SRV-8-infected monkeys. (**A**) Heatmap for differentially expressed genes in the PBMCs between the healthy and five SRV-8-infected monkey groups (*p* < 0.05; log2(FC) > 2 or log2(FC) < −2). The heat map is representative of the mean DEG of several samples in each group. (**B**) The number of genes that upregulated and downregulated in the SRV-infected monkey groups when compared with the healthy monkey group (P^−^V^−^A^−^).

**Figure 5 viruses-15-01538-f005:**
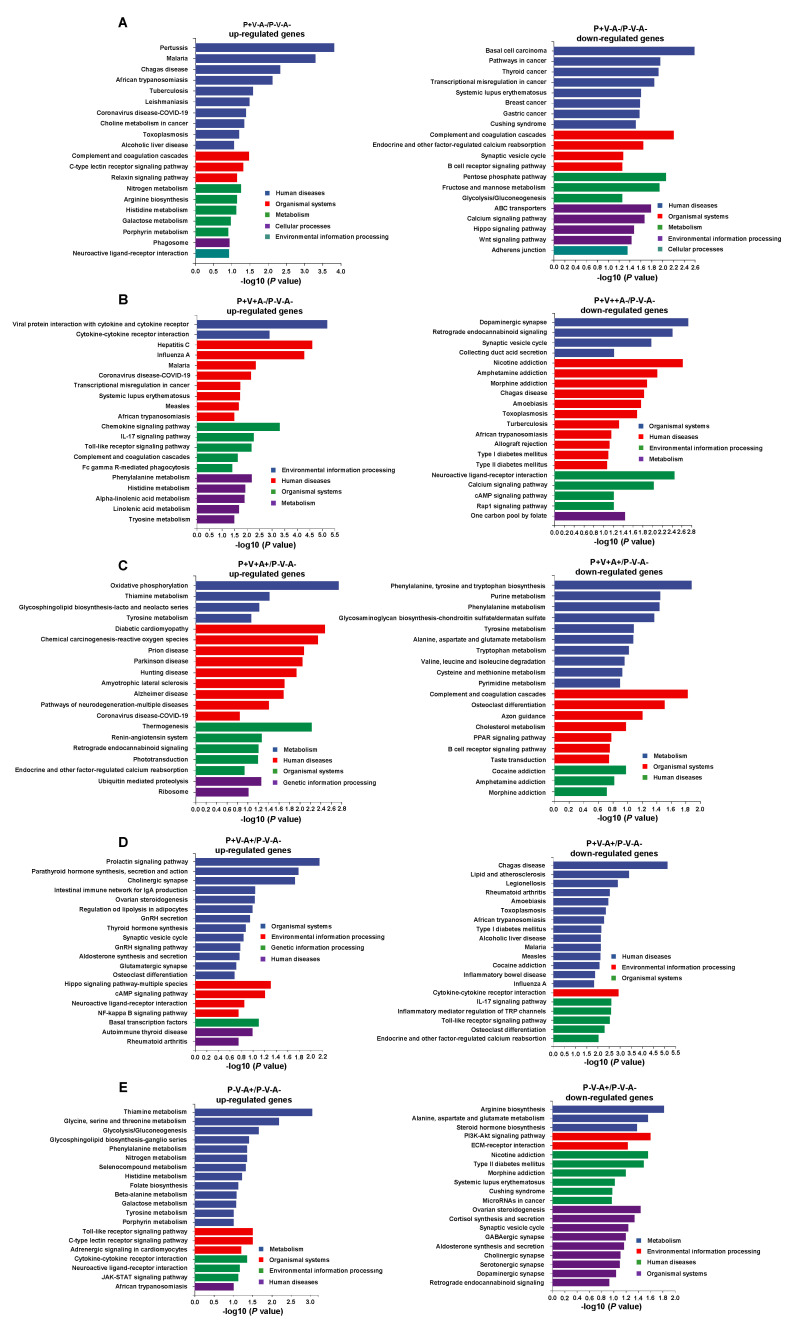
DEG-based KEGG enrichment analyses in the PBMCs of monkey groups P^+^V^−^A^−^ (**A**), P^+^V^+^A^−^ (**B**), P^+^V^+^A^+^ (**C**), P^+^V^−^A^+^ (**D**), and P^−^V^−^A^+^ (**E**). The DEGs with fold change ≥ 4 (*p* < 0.05) were used. The up- and downregulated genes were analyzed separately.

**Table 1 viruses-15-01538-t001:** Oligonucleotides used in this study.

Oligonucleotides Name	Sequence (5′-3′)	Description
SRV identification		
*SRV-1*-F	GAGACAAATCTCCCTCCAGTGGTGACG	Forward primer for *SRV-1* provirus
*SRV-1*-R	CAGCAATCTTCGGCTAATTGGCGTTGG	Reverse primer for *SRV-1* provirus
*SRV-2*-F	GCGAGGCTATTACTATGATACACCAGC	Forward primer for *SRV-2* provirus
*SRV-2*-R	CTTTAGGATTCCAGCAAATGGGCTGACC	Reverse primer for *SRV-2* provirus
*SRV-4*-F	GTTAGTAATGTTCCTACAGTCATAGGATCAGG	Forward primer for *SRV-4* provirus
*SRV-4*-R	GGTAAGGTTTTCATAGGTGTAATTACTGGGTAAGG	Reverse primer for *SRV-4* provirus
*SRV-5*-F	GCAATGGTACAACTTATAATACAGCTAAATTGC	Forward primer for *SRV-5* provirus
*SRV-5*-R	GTCAGATGCATTGGCCAAAGATAAATTTTGG	Reverse primer for *SRV-5* provirus
*SRV-8*-F	CAGCTTACTCCCAAGTAGGTTCC	Forward primer for *SRV-8* provirus
*SRV-8*-R	GAGATAGATATGTTATAAGTGTTAGTACAATTGG	Reverse primer for *SRV-8* provirus
*GAPDH*-F	GGATATTGTTGCCATCAATGACC	Forward primer for *GAPDH*
*GAPDH*-R	CCTTCTCCGTGGTGGTGAAGAC	Reverse primer for *GAPDH*
qPCR		
qRT-mk-*GAPDH*-F	GAGTCCACTGGCGTCTTCA	Forward qRT-PCR primer for *GAPDH*
qRT-mk-*GAPDH*-R	TCTTGAGGCTGTTGTCATACTTC	Reverse qRT-PCR primer for *GAPDH*

## Data Availability

The RNA-Seq raw data generated in the present study were deposited in the SRA under accession number PRJNA946895.

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
