# Peer review of "Characterizing the Pathogenicity and Immunogenicity of Simian Retrovirus Subtype 8 (SRV-8) Using SRV-8-Infected Cynomolgus Monkeys"

_viruses, 2023, doi:10.3390/v15071538_

Round 1

Reviewer 1 Report

See attached file

Reviewer 2 Report

The manuscript described the physiopathology of simian retrovirus infection SRV of a monkey cohort. The authors identified 120 animals infected by the SRV-8 phenotype, and after stratification according to symptoms they analyzed immune cell populations and transcriptomic profiles with the aims to correlate infection status to symptoms.

The mansucript is well written and provide interesting results. However several major point need to be addressed.

1) In the introduction it would be worth noting which cells are targetted by SRV-8 as general readers might not be familiar with this specific virus. Moreover the cell tropism might also help to support their finding.

2) In fig 1, the legend is missing for panel E.

3) In Fig 2C scale bar are missing. the figure is not convincing to support the statement that Raji B cells are infected : discrimination of infected donors cells from Raji receiver cells must be provided (viral staining would help or at least staining of Raji cells with labelling dye). Moreover, indication of infected Raji cells, by viral staining is mandatory  to state that they are infected. Cytopathology is not enough to support infection and whether cytopathology targets Raji receiver cells or infected donor cells can be assessed in this experiment.

4) In fig 3A changes in RBD and HBG are modest although significant.  What is the relevance of these low difference ?

What are the markers used  for defining helper T cells, CD8 or HSC ? The gating strategy. must be shown. Naming of panel is confusing, for example panel B has 2 graphs and panel C has 4 but their agencement is not trivial.

5) In Fig4A, is the heat map representative of the mean DEG of several samples in each group ?

The return to homeostasis in the P-V-A+ group can not be stated only by the decreased number of DEG. The nature of the genes is important, and the heat map shows a very different profile in P-V-A+ compared to non infected. Return to homeostasis must be documented further by comparing specific genes and not the global number.

6) Figure 5 is difficult to read due to too small text

Author Response

Please see the attachment~

Round 2

Reviewer 1 Report

The authors have addressed most points. However, the following should be mentioned:

1. The explanation for only using female primates should be included in the final paper.

2. Was there a reason for not including SRV-3 primers in the PCR?

3.I am still not happy with the description of animal numbers used in the study. This has to be made crystal clear in the light of ethical concerns of using animals for research.

They started with 120 animals. Of these, 92 were unstable during the tracking process. What does this mean exactly?

Of the 28 carried forward for the study, 5 had the P-V-A- phenotype. This should be shown. This leaves a study number of 23. 

Please explain this clearly in the text before publication.

In the authors' letter, they talk of "the other16". Please clarify this.

Once these points have been addressed and the text is unambiguous, the paper can be accepted for publication.

The English in the authors' letter was not of the same standard as the paper. Please ensure when transferring information from the rebuttal letter to the modified paper that the English is clear.

Author Response

Dear Reviewer:
